# Incidence, Risk Factors, and Outcomes of Symptomatic Bone Cement Displacement following Percutaneous Kyphoplasty for Osteoporotic Vertebral Compression Fracture: A Single Center Study

**DOI:** 10.3390/jcm11247530

**Published:** 2022-12-19

**Authors:** Junbo Qi, Yuanyu Hu, Zhongwei Yang, Yanlei Dong, Xin Zhang, Guojin Hou, Yang Lv, Yan Guo, Fang Zhou, Bingchuan Liu, Yun Tian

**Affiliations:** 1Department of Orthopedics, Peking University Third Hospital, No. 49 North Garden Road, Haidian District, Beijing 100191, China; 2Engineering Research Center of Bone and Joint Precision Medicine, No. 49 North Garden Road, Haidian District, Beijing 100191, China; 3Beijing Key Laboratory of Spinal Disease Research, No. 49 North Garden Road, Haidian District, Beijing 100191, China; 4Information Management and Big Data Center, Peking University Third Hospital, No. 49 North Garden Road, Haidian District, Beijing 100191, China

**Keywords:** osteoporotic vertebral compression fracture, percutaneous kyphoplasty, bone cement displacement, incidence, risk factor, outcome

## Abstract

Study design: Retrospective. Background: Symptomatic bone cement displacement (BCD) is a rare complication following percutaneous kyphoplasty (PKP) interventions for osteoporotic vertebral compression fracture (OVCF). This study aimed to investigate the incidence and the outcomes of symptomatic BCD comprehensively and identify its risk factors. Methods: The clinical data of patients treated with PKP for OVCF between January 2012 and December 2020 were extracted. Patients who developed BCD following PKP during follow-up were divided into the symptomatic and asymptomatic groups. Patients who did not develop BCD were assigned to the control group. Univariate and multiple logistic regression analyses were used to compare the three clinical groups’ features to assess the independent risk factors for the symptomatic and asymptomatic groups. Results: A total of 896 patients were enrolled. Twenty-one patients (2.3%) were identified as having symptomatic BCD following PKP for OVCF, and 35 (3.9%) developed asymptomatic BCD. Compared with the control group, the symptomatic and asymptomatic groups had a higher incidence of anterior leakage, intravertebral vacuum cleft (IVC) signs, and a lower cement distribution score. The symptomatic group had a lower relative cross-sectional area (rCSA) of the paraspinal muscle (PSM), higher PSM fatty degeneration, and higher kyphotic angle (at the last follow-up) than the asymptomatic and control groups. For outcomes, the symptomatic group had a higher VAS/ODI score and a higher incidence of new vertebral fractures compared with the asymptomatic and control groups. Anterior leakage (OR: 1.737, 95% CI: 1.215–3.300), the IVC sign (OR: 3.361, 95% CI: 1.605–13.036), the cement distribution score (OR: 0.476, 95% CI: 0.225–0.904), PSM rCSA (OR: 0.953, 95% CI: 0.917–0.992), and PSM fatty degeneration (OR: 1.061, 95% CI: 1.005–1.119) were identified as independent risk factors for the symptomatic group. Anterior leakage (OR: 1.839, 95% CI: 1.206–2.803), the IVC sign (OR: 2.936, 95% CI: 1.174–9.018), and cement distribution score (OR: 0.632, 95% CI: 0.295–0.858) were independent risk factors for the asymptomatic group. Conclusion: The incidence of symptomatic BCD is 2.3% in patients treated with PKP. Anterior leakage, the IVC sign, and the distribution score were independent risk factors for BCD, and paraspinal muscle degeneration was a specific risk factor for symptomatic BCD. Symptomatic BCD can lead to poor outcomes.

## 1. Introduction

Percutaneous kyphoplasty (PKP) and vertebroplasty (PVP) are major, minimally invasive surgical procedures used to treat osteoporotic vertebral compression fractures (OVCF) [1,2,3]. Specifically, their efficacy in treating OVCF has been widely acknowledged because of their prominent advantages, including immediate pain relief, reduced invasiveness, restoration of vertebral height, and prevention of deformity and deterioration [2,3,4].

However, extensive use of PKP can lead to risks and complications, such as bone cement leakage [5], puncture site bleeding [6], partial or complete paraplegia [7], transient hypotensive response [8], and even delayed infection or refracture [9,10]. Bone cement displacement (BCD) is also a rare complication and should not be overlooked. BCD can be caused by acute trauma, infection, cement loosening, and breakage and may lead to severe symptoms [11,12,13]. Conservative treatment is ineffective for patients with symptomatic bone cement displacement after PKP, and revision surgery might become challenging due to advanced age and multiple comorbidities. Therefore, a comprehensive understanding of symptomatic BCD is needed to ensure that an optimal opportunity for precaution and intervention is not missed until revision surgery becomes unavoidable.

Currently, most studies on BCD are case reports. Only Gao et al. [14] have explored the potential independent risk factors associated with BCD. However, their study only focused on radiographic displacement and did not provide an etiological explanation for the severe clinical symptoms. Furthermore, studies on the incidence and outcomes of BCD are still lacking. To provide additional clinical data and comprehensive insights on BCD, we investigated the incidence and outcomes of symptomatic BCD in the present study and identified the risk factors in patients who underwent PKP treatment for OVCF. In addition, we enrolled asymptomatic patients with BCD in our study and compared the clinical characteristics between symptomatic and asymptomatic BCD to define the incidence of symptoms.

## 2. Materials and Methods

### 2.1. Subjects

Figure 1 shows the inclusion and exclusion criteria and details of the study selection process. A total of 896 patients with OVCF who underwent PKP treatment at our hospital between January 2012 and December 2020 and a routine 6-month follow-up were included. The patients underwent radiographic examinations at the follow-up. Of the enrolled patients, 56 patients were diagnosed with BCD by radiography during follow-up, and the remaining 840 patients were assigned to the control group. Of the 56 patients with BCD, 21 were further assigned to the symptomatic and 35 to the asymptomatic groups. Based on expert recommendations, the criterion for radiographic diagnosis of BCD was >2 mm movement of the anterior edge of the cement at follow-up compared with before discharge, as referenced by taking the anterior wall of the vertebral body (Figure 2). The criteria for symptomatic patients were progressive worsening of back pain, an increase of ≥3 points in the Visual Analogue Scale (VAS) score at follow-up compared to that at discharge, or a newly presented neurologic deficit, whereas the criterion for asymptomatic patients only included a radiographic diagnosis of BCD. All patients were instructed to wear a brace for at least one month. This study was approved by the Medical Science Research Ethics Committee of our institution (reference No. M2021273) and was accomplished following the Strengthening the Reporting of Observational studies in Epidemiology (STROBE) statements (Appendix A).

### 2.2. Data Collection

Clinical data were extracted from the electronic medical records, including demographic information, vertebral distribution, cement leakage, the volume of bone cement bilaterally injected, VAS and Oswestry Disability Index (ODI) scores (at admission/discharge), bracing time, and use of osteoporosis medication. The VAS/ODI scores (at the last follow-up), the presence of new vertebral fractures, and details on the treatment of symptomatic patients with BCD were also collected for outcome evaluation.

Radiological parameters were also measured. The intravertebral vacuum cleft (IVC) sign was defined as a half-moon-shaped euphotic area located in the fractured vertebral body on plain radiography or a low-density area on computed tomography (CT) [15]. The height of the affected vertebral body was measured at the point of maximum vertebral collapse. The collapse of the vertebral (%) was measured using the following formula [16]: [(upper vertebral height + lower vertebral height)/2 − affected vertebral height]/[(upper vertebral height + lower vertebral height)/2] × 100 (Figure 3). The preoperative, postoperative, and last follow-up kyphotic angle was measured using the Cobb method (Figure 3), and the restoration of the kyphotic angle (%) was measured as follows: (preoperative kyphotic angle − postoperative kyphotic angle)/preoperative kyphotic angle. The preoperative L1 Hounsfield unit (HU) was measured using the Picture Archiving and Communication System (PACS), as described in a previous study [17].

Measurement of relative cross-sectional area (rCSA) and paraspinal muscle (PSM) fatty degeneration was based on the T2-weighted axial magnetic resonance imaging (MRI) slice at the level of the inferior vertebral endplate of L4. Images were processed using Image J software (version 1.53, U.S. National Institutes of Health, Bethesda, MD, USA). The total cross-sectional area (tCSA) of the paraspinal muscle and the intervertebral disc was measured using the region of interest (ROI), and the functional cross-sectional area (fCSA) of the paraspinal muscle was measured using the threshold method (Figure 4) [16,18,19]. The rCSA was defined as the ratio of muscle tCSA to disc tCSA and was used to balance differences in body size [19]. Fatty degeneration of the paraspinal muscle was defined as the ratio of fCSA to tCSA [16].

Measurement and calculation of the bone-cement distribution score were based on the method described by Liu et al. [20]. The vertebra was divided into quadrants on X-ray films based on the anteroposterior and lateral positions. If bone cement filling exceeded one-third of the quadrant, the quadrant was counted as one point. If bone cement contact was detected on the upper or lower endplate in the lateral position, each contact counted as a point. The total possible score was 10. Two cases with different distribution scores are shown in Figure 5.

### 2.3. Review of the Literature

Relevant literature was systematically reviewed from 2000 to 2022 using the PubMed and Web of Science databases using a combination of Boolean operators with the following subject headings and keywords: vertebroplasty, kyphoplasty, cement displacement, cement dislodgment, cement extrusion, vertebral collapse, complications, and revision treatment. The results were limited to articles in English. Due to the rarity of cases after PKP, all eligible reports and clinical series published on BCD after percutaneous surgery were assembled.

### 2.4. Statistical Analyses

First, univariate analysis was used to compare clinical characteristics and outcomes between each group and to identify potential risk factors. The Shapiro–Wilk test was used to assess the normality of the data. A one-way analysis of variance (ANOVA) and the least significant difference (LSD) post-hoc test were used for continuous, normally distributed variables. The Kruskal–Wallis and Dunn post-hoc tests were used for continuous, non-normally distributed variables. Chi-square and post-hoc tests were used for categorical variables. These potential risk factors with statistical significance (*p* < 0.05) in the univariate analysis were then subjected to multiple logistic regression analyses to identify independent risk factors between groups, and the adjusted odds ratios (OR) with 95% confidence intervals (CI) were calculated. To test reliability, the radiological parameters of 100 randomly selected patients were independently measured by two observers, and the intraclass correlation (ICC) was calculated. An ICC of > 0.75 indicated good reliability [21]. SPSS version 27.0 (Armonk, NY, USA) was used for all data analyses, and the level of significance was set at *p* < 0.05.

## 3. Results

A total of 896 patients with complete clinical data were enrolled (72.21 ± 9.22 years, 76.5% female). Among them, 56 (6.2%) were diagnosed with BCD, including 21 (37.5%) in the symptomatic group and 35 (62.5%) in the asymptomatic group. The remaining 840 patients were assigned to the control group. The overall incidence of symptomatic BCD was 2.3%. Most of the fractures occurred in the thoracolumbar segment (83.4%, Figure 6). All patients underwent PKP surgery with a bilateral injection of bone cement (5.12 ± 1.01 mL). After PKP, the kyphosis deformity improved, and the VAS/ODI was significantly lower at discharge than at admission in all three groups (*p* < 0.05). Table 1 shows that all measurements taken from radiographs were in good agreement (>0.75) with inter-observer reliability.

### 3.1. Univariate Analysis

Compared to the control group, the symptomatic and asymptomatic groups showed a higher incidence of anterior cement leakage (52.4 vs. 65.7 vs. 18.7%) and a higher incidence of IVC signs (14.3 vs. 11.4 vs. 3.8%). Similarly, the bone cement distribution score of the symptomatic and asymptomatic groups was significantly lower than that of the control group (7.73 ± 1.88 vs. 7.85 ± 1.76 vs. 8.08 ± 1.43). The degree of paraspinal muscle degeneration in the symptomatic group was significantly higher than that in the asymptomatic and control groups, showing significantly lower rCSA and higher paraspinal muscle fatty degeneration. The kyphotic angle in the symptomatic group (21.51 ± 6.19°) was significantly higher than that in the asymptomatic and control groups at the last follow-up. Anterior leakage, IVC sign, cement distribution score, PSM rCSA, PSM fatty degeneration, and kyphotic angle (at the last follow-up) were identified as potential risk factors and subjected to multiple logistic regression analysis for the symptomatic group. For the asymptomatic group, anterior leakage, IVC signs, and the cement distribution score were subjected to the multiple logistic regression analysis. The results are presented in Table 2.

A comparison of the results between groups is presented in Table 3. Compared to the asymptomatic and control groups, the symptomatic group had worse outcomes after the completion of follow-up. In the symptomatic group, the VAS score was 5.47 ± 1.53, and the ODI score was 34.89 ± 8.42 in the symptomatic group, which were significantly higher than those of the other two groups. The incidence of new vertebral fractures was also significantly higher in the symptomatic group than in the control group (42.9 vs. 21.2%). Furthermore, we evaluated the treatments in the symptomatic group, and most of the patients (76.1%) underwent revision surgery to relieve symptoms.

### 3.2. Multiple Logistic Regression Analysis

Multiple logistic regression analysis (Table 4) showed that the independent risk factors of the symptomatic group were anterior leakage (adjusted odds ratio (OR): 1.737, 95% confidence interval (CI): 1.215–3.300), the IVC sign (adjusted OR: 3.361, 95% CI: 1.605–13.036), bone cement distribution (adjusted OR: 0.476, 95% CI: 0.225–0.904), PSM rCSA (adjusted OR: 0.953, 95% CI: 0.917–0.992), and PSM fatty degeneration (adjusted OR: 1.061, 95% CI: 1.005–1.119).

For the asymptomatic group, only anterior leakage (adjusted OR: 1.839, 95% CI: 1.206–2.803), the IVC sign (adjusted OR: 2.936, 95% CI: 1.174–9.018), and bone cement distribution (adjusted OR: 0.632, 95% CI: 0.295–0.858) were independent risk factors.

### 3.3. Review of the Literature

After removing duplicates, screening the title and abstract, and full-text assessment, 14 articles were ultimately included in the final analysis [11,12,13,14,22,23,24,25,26,27,28,29,30,31]. Eighteen patients with BCD have been reported in the literature over the past 20 years, and details of these cases are shown in Table 5. In addition, Gao et al. [14] reviewed the records of 1538 patients with OVCF treated with PVP or PKP from 2016 to 2021 and showed that high restoration of the Cobb angle, cement leakage (anterior edge), limited brace wearing time, and non-postoperative osteoporosis treatment were risk factors of BCD after percutaneous vertebral augmentation. Unlike other case reports, this study was the only systematic retrospective study of BCD to date.

## 4. Discussion

Symptomatic bone cement displacement is a rare complication after PKP for osteoporotic vertebral compression fracture and can lead to worsening pain or newly-presented neurologic deficits and even inevitable revision surgery. However, studies on the systematic analysis of BCD, apart from cases that occur under specific conditions, are still scarce. Gao et al. [14] conducted a study on the risk factors for BCD after percutaneous vertebral augmentation, which is the only systematic retrospective study to date. In our study, we analyzed data from 896 patients treated with PKP for OVCF between January 2012 and December 2020. The overall incidence was 2.3% in patients with OVCF treated with PKP. Among patients with BCD, 37.5% presented with symptomatic BCD. Anterior leakage, IVC sign, and bone cement distribution score were identified as independent risk factors for BCD. Paraspinal muscle degeneration was also found to be an independent risk factor only for symptomatic BCD. Furthermore, symptomatic BCD resulted in worse outcomes and possibly required revision surgery to alleviate symptoms.

### 4.1. Incidence

BCD is a relatively rare PKP complication. In our review, most of the case reports were single case reports, and no incidence of BCD has been reported after PKP. According to a recent study by Gao et al. [14], among 1500 patients with percutaneous vertebral augmentation, 78 developed BCD, with an incidence of approximately 5%. Similar to Gao et al. [14], 56 of the 896 patients (6.2%) were diagnosed with BCD in the present study. However, we divided patients with BCD into symptomatic and asymptomatic groups in contrast to the study by Gao et al. [14], which focused only on patients with imaging findings of BCD. Among patients with BCD, 37.5% exhibited symptoms, representing 2.3% of the total population. In clinical practice, we pay more attention to patients with obvious symptoms and less to asymptomatic patients, who generally do not need to be treated. Through this grouping, we can further clarify the risk factors for symptomatic BCD and identify patients who require treatment.

### 4.2. Risk Factors

In our study, anterior cement leakage was found to be an independent risk factor for BCD. Ding et al. [32] determined that cortical leakage of bone cement was closely related to vertebral cortical disruption. Anterior cement leakage is a sign of damage to the anterior wall of the vertebral body where bone cement is more likely to break through and cause displacement, as mentioned in a previous case report [20]. In the study by Gao et al. [14], anterior leakage was also found to be an independent risk factor, which is similar to our results. Additionally, the IVC sign was another significant factor. Similar to cortical disruption, IVC signs are often observed in patients with BCD, similar to cortical disruption [11,12,25]. IVC may be indicative of bone nonunion with dynamic instability and avascular necrosis of the fractured vertebra [33,34]. Once IVC occurs, the injected cement may present less interdigitation with the surrounding bone due to the existence of a cystic cavity, making the cement simply a space-occupying material without mechanical integration with the cancellous structure of the surrounding host, increasing the risk of displacement [35].

Another independent risk factor for BCD is the bone cement distribution score. The degree of bone-cement distribution is closely related to the complications of OVCF [36]. Current studies believe that a good distribution of bone cement is indicative of better diffusion and integration of cement within the trabecular microstructures [37], providing stress balance and ensuring better immobility. Previously, the types of distribution and the volume of bone cement were commonly used as distribution parameters [38]. However, these parameters may not be the best choice considering differences in vertebral size and morphology [39,40]. Many studies have shown that the volume fraction of bone cement calculated based on CT and the cement distribution model obtained by computer modeling may be more accurate [41,42]. Gao et al. [14] developed a new parameter (interweaving degree of bone cement) that is measured using a three-dimensional finite element method to evaluate the distribution and found it to be an independent risk factor for BCD. However, these methods are too complicated to be used in clinical practice. Additionally, compared to CT, X-rays deliver a small dose of radiation and are relatively more cost-effective, which is generally used at routine follow-ups. We selected an X-ray measurement method to evaluate bone cement distribution, which was invented and improved by Liu et al. [20]. It has been proven to be effective in predicting the risk of new fractures and recollapse and is suitable for evaluating bilaterally distributed cement. In the present study, a low distribution score may indicate a worse distribution of bone cement and loose binding between bone cement and the trabecular microstructure, which is more likely to lead to the appearance of BCD. However, from another perspective, a better distribution is closely related to the appearance of bone cement leakage [36]. In conclusion, the proper degree of cement distribution helps reduce the incidence of complications.

Compared with previous studies, our study is the first to investigate the effect of paraspinal muscle degeneration on BCD. Interestingly, paraspinal muscle degeneration is an independent risk factor unique to symptomatic BCD and can partly explain the appearance of symptoms. Paraspinal muscle degeneration is an early clinical manifestation of sarcopenia [43,44], which is widely observed in older adults and is characterized by decreased muscle strength and quality of life [45]. PSM rCSA and PSM fatty degeneration are common indicators of paravertebral muscle degeneration [16,19]. PSM rCSA is a predictor of sarcopenia and can reflect the relative volume of the paraspinal muscles. Low rCSA is an independent risk factor for complications (refracture and pain) in patients with OVCF after PKP [46,47]. As a supplement to rCSA, fatty degeneration reflects muscle mass and functional levels. A high level of fatty degeneration in PSM indicates a decrease in the support and maintenance function of the paraspinal muscles and is associated with a poor prognosis in patients with OVCF, such as rCSA [16].

In the present study, a higher rCSA and a lower level of PSM fatty degeneration were found in the symptomatic group than in the asymptomatic group. This may indicate that the mass and function of the paraspinal muscles were better in the asymptomatic group than in the symptomatic group. After PKP, the cement-reinforced vertebrae and adjacent vertebrae form a strength gradient [48]. If the mass and strength of the paraspinal muscles cannot compensate for the effect of local compressive changes, structural damage may occur, such as recollapse [49]. Based on this mechanism, we hypothesized that paravertebral muscles provide support and tension to the anterior column of the vertebral body to some extent [18], limiting the progression of anterior cement displacement and progressive kyphosis. Early loss of correction and rekyphosis are common complications after surgical treatment for OVCF, even in patients with percutaneous fixation of the pedicle screws [50]. Degeneration of paraspinal muscles may aggravate this loss of correction and lead to recollapse and clinical symptoms. As shown in our study, the symptomatic group had a higher mean kyphotic angle (21.51 ± 6.19°) at the last follow-up, which could be explained by the degeneration of the paravertebral muscle that led to its loss of function, resulting in progressive kyphosis and spinal cord compression and leading to severe clinical symptoms.

In addition, there is a muscle–bone interaction, which shows that muscles can support the mechanical strength of bones and maintain the function of the musculoskeletal system [51]. A previous study showed that paraspinal muscle degeneration could cause the loss of adjacent bone [52]. Another hypothesis is that paraspinal muscle degeneration leads to trabecular damage, causing a decrease in bone–cement binding. Cortical thinning makes it easier for bone cement to break through the cortex and cause severe displacement. However, further research is required to confirm this hypothesis. Based on the above possibility, proper exercise training, physical therapy, and good nutrition should be recommended after PKP, which may effectively improve the condition of the paraspinal muscles and avoid the occurrence of kyphosis and symptomatic BCD.

### 4.3. Outcomes

Systematic prognostic studies that involve patients with BCD are lacking. Previous case reports have suggested that patients with symptomatic BCD have a poor prognosis. Of the 18 patients reviewed, all patients had recently presented with worse back pain, and 14 patients (77.7%) underwent a second surgery as a treatment strategy [11,12,13,22,23,24,25,26,27,28,29,30,31]. The results of our study were similar to those reported in previous studies. The results of the symptomatic group were significantly worse than those of the asymptomatic and control groups, indicated by an increase in the VAS/ODI score, an increased incidence of refractures and requiring revision surgery.

Conservative treatment is acceptable for symptomatic patients. However, if symptoms of vascular and nervous compression or progressive kyphosis are confirmed, then subsequent revision surgery should be recommended more aggressively [53,54]. During the revision surgery process, the augmentation of pedicle screws with PMMA cement was implemented to improve initial fixation and fatigue strength. Then, thorough decompression of the nerve and spinal cord can be implemented by further laminectomy. After confirming successful decompression, the fixation devices were carefully assembled, and autologous bone was implanted in the intertransverse region to promote fusion. In this way, the normal spinal sequence and mechanical stability can be reconstructed. Based on our clinical experience, prompt revision surgery for BCD always results in symptomatic relief. We believe that revision surgery is recommended for symptomatic relief in the absence of contraindications.

### 4.4. Limitations

This study had several limitations. The retrospective nature of this study leads to difficulties in data availability and bias control. Additionally, the rarity of BCD incidences results in a relatively small sample size, and it is difficult to conduct prospective studies to verify independent risk factors. Third, studies on BCD are mostly case reports; thus, as mentioned above, systematic studies are still lacking.

## 5. Conclusions

Our study revealed that, among patients with BCD, 37.5% had symptomatic BCD, and the overall incidence is 2.3% in patients with OVCF treated with PKP. Additionally, anterior leakage, IVC sign, and bone cement distribution score are identified as independent risk factors for BCD. Paraspinal muscle degeneration is also determined to be a risk factor only for symptomatic BCD, which may contribute to the occurrence of symptoms, progression of BCD, and kyphosis deformity. Symptomatic BCD results in worse outcomes and therefore requires early identification and intervention in clinical practice.

## Figures and Tables

**Figure 1 jcm-11-07530-f001:**
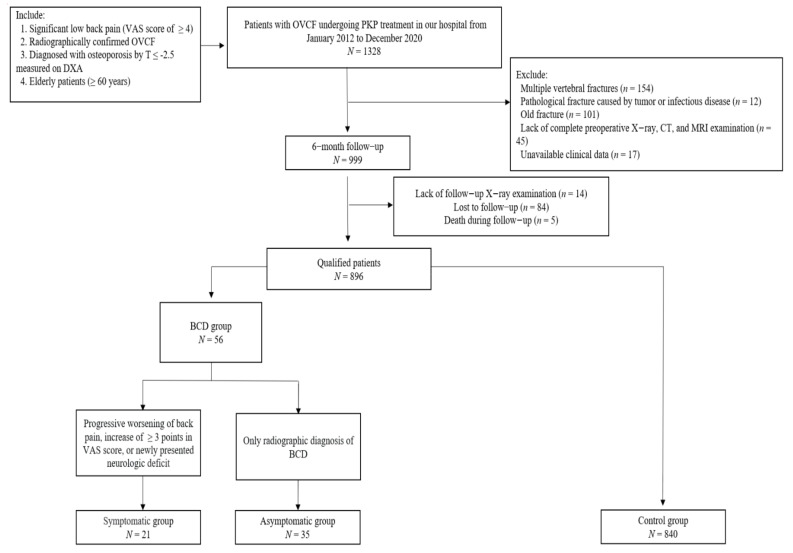
The flowchart of the study. OVCF = osteoporotic vertebral compression fracture, PKP = percutaneous kyphoplasty, DXA = dual-energy X-ray absorptiometry, CT = computed tomography, MRI = magnetic resonance imaging, BCD = bone cement displacement, and VAS = Visual Analogue Scale.

**Figure 2 jcm-11-07530-f002:**
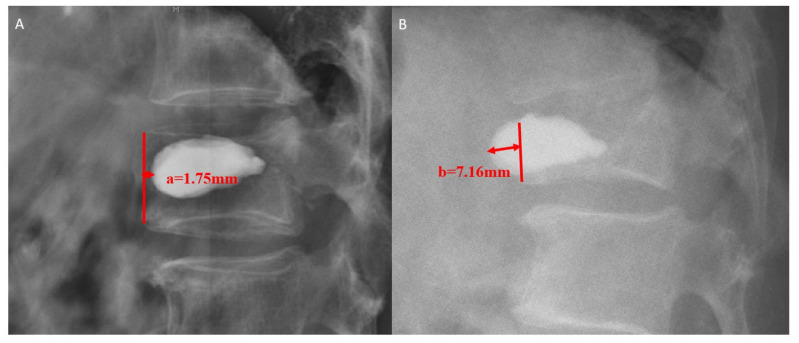
(**A**) The patient showed well-located bone cement, and the distance between the cement’s anterior edge and the anterior wall of the vertebral body was 1.75 mm before discharge. (**B**) The patient showed that the distance was 7.16 mm at follow-up. By taking the anterior wall of the vertebral body as a reference, the movement of the anterior edge of bone cement was 8.91 mm, and the patient met the criteria for radiographic diagnosis of BCD.

**Figure 3 jcm-11-07530-f003:**
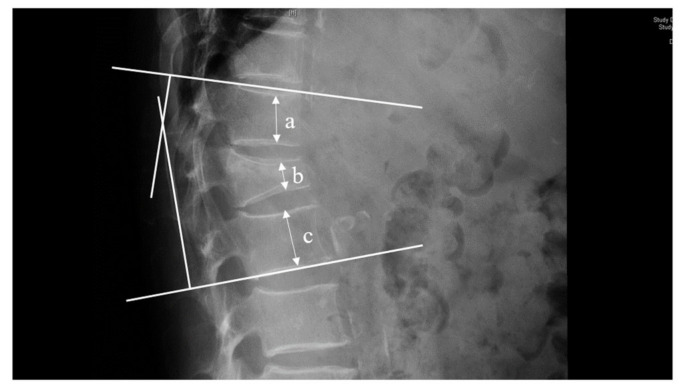
Measurement of vertebral collapse and kyphosis. Vertebral collapse (%) was measured as follows: [(a + c)/2 − b]/ [(a + c)/2] × 100. The kyphosis Cobb was measured by using the Cobb method. a = upper vertebral height, b = affected vertebral height, c = lower vertebral height.

**Figure 4 jcm-11-07530-f004:**
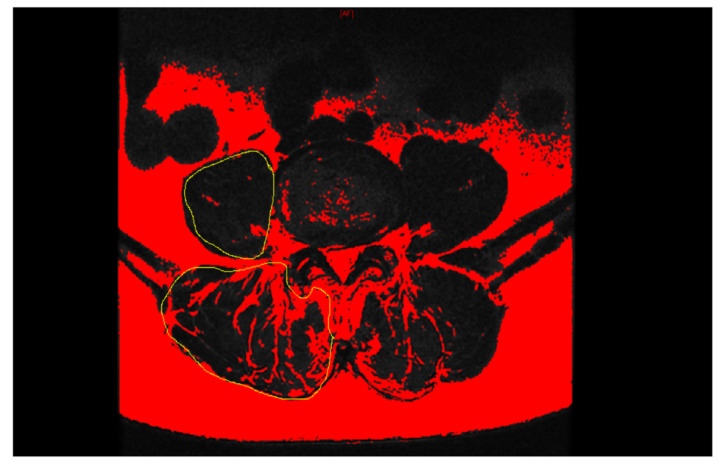
The tCSA of PSM was measured by drawing the outline of the fascial–muscle boundary using the ROI at the level of the inferior vertebral endplate of L4 on axial T2-weighted MRI. Measurement of fCSA of PSM was performed using a threshold method. tCSA = total cross-sectional area, PSM = paraspinal muscle, ROI = region of interest, and fCSA = functional cross-sectional area.

**Figure 5 jcm-11-07530-f005:**
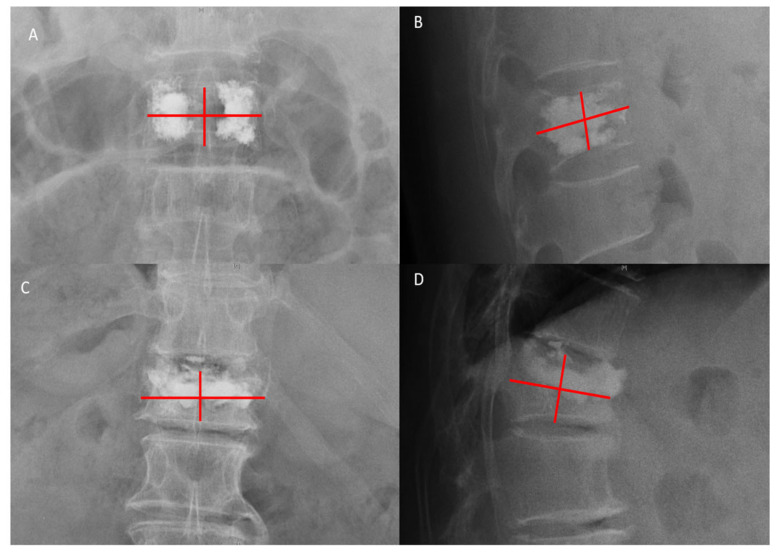
(**A**,**B**) A 74-year-old male patient with OVCF at T12. Anteroposterior and lateral images were taken after PKP, and the bone cement distribution score was calculated as 4 + 4 + 2 = 10. (**C**,**D**) A 77-year-old female patient with OVCF at T12. The bone cement distribution score was calculated as 2 + 2 + 1 = 5.

**Figure 6 jcm-11-07530-f006:**
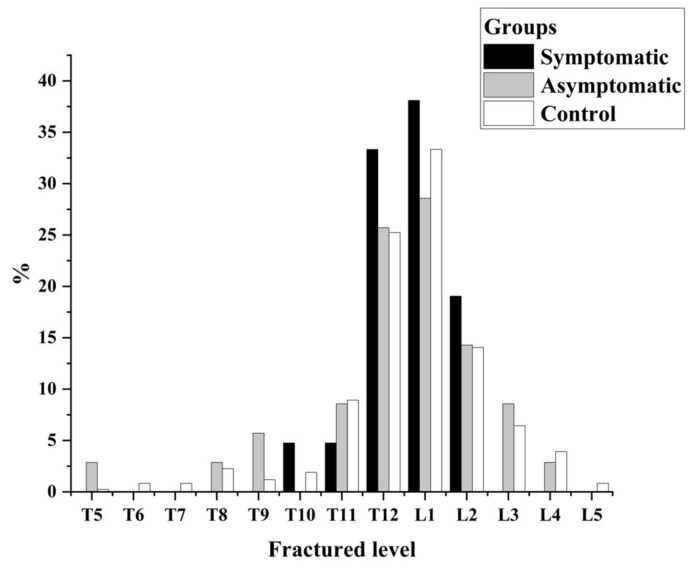
The proportion of fractured levels in the symptomatic, asymptomatic, and control groups.

**Table 1 jcm-11-07530-t001:** Inter-observer reliability for each measurement.

	ICC	95% CI	*p*
Radiographic diagnosis of BCD	0.944	0.910–0.957	<0.001
IVC sign	0.951	0.939–0.982	<0.001
Vertebral collapse	0.776	0.704–0.841	0.002
L1-HU	0.889	0.874–0.902	<0.001
PSM rCSA	0.818	0.715–0.934	<0.001
PSM fatty degeneration	0.783	0.756–0.807	<0.001
Kyphotic angle			
Preoperative	0.907	0.854–0.966	<0.001
Postoperative	0.861	0.836–0.904	<0.001
At the last follow-up	0.881	0.865–0.894	<0.001
Bone cement distribution score	0.814	0.790–0.835	<0.001

ICC, Intraclass correlation coefficient; CI, confidence interval; BCD, bone cement displacement; HU, Hounsfield unit; PSM, paraspinal muscle; rCSA, relative cross-sectional area.

**Table 2 jcm-11-07530-t002:** Intergroup comparison of patients’ characteristics.

*N* (%)	Symptomatic Group (a) *n* = 21 (2.3%)	Asymptomatic Group (b) *n* = 35 (3.9%)	Control Group (c) *n* = 840 (93.8%)	*p* *	*p* _a−c_	*p* _b−c_	*p* _a−b_
Demographic							
Age (years)	71.48 ± 9.63	73.85 ± 10.06	72.39 ± 9.14	0.755	0.763	0.496	0.467
Female (n, %)	16 (76.1)	27 (77.1)	643 (76.5)	0.989	0.974	0.895	0.935
BMI (Kg/m^2^)	24.25 ± 3.10	25.00 ± 3.86	24.23 ± 3.64	0.612	0.978	0.466	0.555
Hypertension (n, %)	7 (33.3)	15 (46.9)	367 (43.7)	0.613	0.344	0.922	0.480
Diabetes (n, %)	2 (9.5)	9 (25.7)	157 (18.7)	0.285	0.285	0.299	0.140
Vertebral distribution				0.304	0.328	0.305	0.168
Thoracic region (n, %)	0 (0)	4 (11.4)	45 (5.9)				
Thoracolumbar region (n, %)	20 (95.2)	27 (77.1)	701 (82.6)				
Lumbar region (n, %)	1 (4.8)	4 (11.4)	94 (11.4)				
IVC sign (n, %)	3 (14.3)	4 (11.4)	32 (3.8)	0.008	0.016	0.026	0.754
Vertebral collapse (%)	32.02 ± 11.15	32.41 ± 12.94	29.61 ± 8.66	0.171	0.212	0.178	0.841
L1-HU (Hu)	78.89 ± 14.71	77.84 ± 12.16	79.82 ± 11.63	0.525	0.717	0.328	0.740
Paraspinal muscle (L4-5)							
PSM rCSA (%)	150.05 ± 14.41	159.88 ± 23.41	159.20 ± 17.42	0.043	0.023	0.825	0.046
PSM fatty degeneration (%)	38.52 ± 12.41	32.33 ± 8.27	34.13 ± 8.23	0.028	0.020	0.213	0.008
Kyphotic angle							
Preoperative (°)	22.06 ± 5.87	21.71 ± 6.20	21.10 ± 3.34	0.313	0.203	0.519	0.869
Postoperative (°)	20.75 ± 7.31	18.39 ± 5.19	18.31 ± 5.83	0.157	0.076	0.953	0.054
Restoration of kyphotic angle (%)	17.21 ± 11.63	20.79 ± 14.34	23.38 ± 17.71	0.207	0.113	0.393	0.339
At the last follow-up (°)	21.51 ± 6.19	18.72 ± 5.38	18.87 ± 5.75	0.045	0.037	0.879	0.046
Volume of bone cement injected (mL)	5.33 ± 0.73	5.08 ± 0.76	5.12 ± 1.02	0.607	0.323	0.904	0.475
Cement leakage				<0.001	0.002	<0.001	0.588
No leakage (n, %)	8 (38.1)	9 (25.7)	571 (68.0)				
Anterior leakage (n, %)	11 (52.4)	23 (65.7)	157 (18.7)				
Others (n, %)	2 (9.5)	3 (8.6)	112 (13.3)				
Bone cement distribution score	7.73 ± 1.88	7.85 ± 1.76	8.08 ± 1.43	<0.001	0.007	0.042	0.824
VAS							
At admission	6.06 ± 1.66	6.11 ± 1.72	6.12 ± 0.98	0.612	0.335	0.932	0.411
At discharge	3.68 ± 1.12	3.50 ± 0.59	3.57 ± 0.57	0.571	0.217	0.960	0.397
ODI							
At admission	47.98 ± 9.31	50.50 ± 5.22	49.52 ± 5.65	0.284	0.216	0.573	0.433
At discharge	21.15 ± 7.46	21.66 ± 5.68	20.96 ± 3.99	0.812	0.835	0.534	0.751
Preoperative drug consumption				0.147	0.096	0.359	0.115
None (n, %)	6 (28.6)	3 (8.6)	126 (15.0)				
NSAIDs (n, %)	10 (47.6)	18 (51.4)	341 (40.6)				
Opioids (n, %)	5 (23.8)	14 (40.0)	373 (44.4)				
Postoperative drug consumption				0.418	0.491	0.304	0.187
None (n, %)	9 (42.9)	21 (60.0)	452 (53.8)				
NSAIDs (n, %)	11 (52.4)	12 (28.6)	332 (39.5)				
Opioids (n, %)	1 (4.8)	4 (11.4)	56 (6.7)				
Brace wearing time				0.582	0.444	0.602	0.362
<1 month (n,%)	5 (23.8)	13 (37.1)	266 (31.7)				
≥1 month (n,%)	16 (76.2)	22 (62.9)	574 (68.3)				
Osteoporosis medication (n, %)	16 (76.2)	26 (74.3)	609 (72.5)	0.775	0.708	0.817	0.873

BMI, body mass index; IVC, intravertebral vacuum cleft; HU, Hounsfield unit; PSM, paraspinal muscle; rCSA, relative cross-sectional area; VAS, Visual Analogue Scale; ODI, the Oswestry Disability Index. * *p* < 0.05 means a significant difference among these groups.

**Table 3 jcm-11-07530-t003:** Intergroup comparison of patients’ outcomes.

*N* (%)	Symptomatic Group (a) *n* = 21 (2.3%)	Asymptomatic Group (b) *n* = 35 (3.9%)	Control Group (c) *n* = 840 (93.8%)	*p* *	*p* _a−c_	*p* _b−c_	*p* _a−b_
VAS	5.47 ± 1.53	3.38 ± 1.23	3.33 ± 0.96	<0.001	<0.001	0.577	<0.001
ODI	34.89 ± 8.42	21.98 ± 7.10	20.66 ± 4.12	<0.001	<0.001	0.148	<0.001
New vertebral fractures (*n*, %)	9 (42.9)	8 (22.9)	178 (21.2)	0.048	0.017	0.813	0.115
Treatment					-	-	-
Reoperation (*n*, %)	16 (76.1)	-	-		-	-	-
Conservative treatment (*n*, %)	5 (23.8)	-	-		-	-	-

VAS, Visual Analogue Scale; ODI, the Oswestry Disability Index. * *p* < 0.05 means a significant difference among these groups.

**Table 4 jcm-11-07530-t004:** Multiple logistic regression analysis.

	Adjusted OR	95% CI	*p*
Symptomatic BCD			
Anterior leakage	1.737	1.215–3.300	0.022
IVC	3.361	1.605–13.036	0.008
Bone cement distribution score	0.476	0.225–0.904	0.025
PSM rCSA	0.953	0.917–0.992	0.017
PSM fatty degeneration	1.061	1.005–1.119	0.009
Asymptomatic BCD			
Anterior leakage	1.839	1.206–2.803	0.013
IVC	2.936	1.174–9.018	0.032
Bone cement distribution score	0.632	0.295–0.858	0.006

BCD, bone cement dislocation; IVC, intravertebral vacuum cleft; PSM, paraspinal muscle; rCSA, relative cross-sectional area.

**Table 5 jcm-11-07530-t005:** Summary of the reported cases suffering symptomatic BCD after PKP or PVP from studies.

Reference	Age (yr)/Sex	Affected Level/ Intervention	Time to BCD	Symptom	Cause	Treatment	FU Time/Outcome
Ha et al. [11]	73/F	T11/PKP	6 weeks	WBP	No trauma	Surgical treatment, fixation T8–L3	2 years/Cured
Tsai et al. [22]	69/M	T12/PVP	1 month	WBP and ND	No trauma	Surgical treatment, fixation T11–L1	NA
Zhang C [12]	73/F	T12/PVP	1 month	WBP and ND	No trauma	Surgical treatment, fixation T10–L2	1 year/BBP but similar muscle strength
Mueller et al. [13]	73/F	L1/PKP	3 weeks	WBP	No trauma	Surgical treatment, fixation T12–L2	NA
Huang et al. [23]	72/M	L1/PVP	30 months	WBP	No trauma	Surgical treatment, fixation T10–L4	>2 years/Cured
Kim JE et al. [24]	75/F	L1/PVP	10 weeks	WBP	No trauma	Surgical treatment, repeat PVP	1 month/BBP
Wagner et al. [25]	75/F	L3/PVP	1 month	WBP	No trauma	Non-surgical treatment	Died soon
Yoshii T et al. [26]	74/F	L3/PVP	1 month	WBP	No trauma	Surgical treatment, fixation T12–S1	1 year/Cured
Shin DA et al. [27]	78/M	L4/PVP	1 month	WBP	No trauma	Non-surgical treatment	NA
Nüchterlein et al. [28]	72/M	L4/PKP	2 months	WBP	Trauma	Surgical treatment, fixation L2–S1	1 year/Cured
Sharma et al. [31]	70/F	L4/PVP	6 months	WBP	No trauma	Non-surgical treatment	NA
Jeong YH et al. [29]	74/F	L4/PKP	1 month	WBP	No trauma	Non-surgical treatment	3 months/BBP but worse kyphosis
Chiu YC et al. [30]	78/M	T11 and T12/PVP	18 months	WBP	No trauma	Surgical treatment, fixation T9–L2	Mean of 1 year/Symptomatic relief
	76/F	T12/PVP	12 months	WBP	No trauma	Surgical treatment, fixation T10–L2	
	69/F	T12/PVP	25 months	WBP	No trauma	Surgical treatment, fixation T9–L4	
	89/M	T12 and L1/PVP	22 months	WBP	No trauma	Surgical treatment, fixation T10–L4	
	71/F	L2 and L3/PVP	9 months	WBP	No trauma	Surgical treatment, fixation T12–L5	
	90/M	L3/PVP	20 months	WBP	No trauma	Surgical treatment, fixation L1–L5	

BCD, bone cement displacement; F, female; M, male; FU, follow-up; PKP, percutaneous kyphoplasty; PVP, percutaneous vertebroplasty; WBP, worse back pain; ND, neurologic deficit; BBP, better back pain; NA, not available.

## Data Availability

The data presented in this study are available upon request from the corresponding author.

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
