# Peer review of "Incidence, Risk Factors, and Outcomes of Symptomatic Bone Cement Displacement following Percutaneous Kyphoplasty for Osteoporotic Vertebral Compression Fracture: A Single Center Study"

_jcm, 2022, doi:10.3390/jcm11247530_

Round 1

Reviewer 2 Report

The aim of the present study was to evaluate the incidence, risk factors, and outcomes of symptomatic bone cement displacement following percutaneous kyphoplasty for osteoporotic vertebral compression fracture. 

Although the idea could be interesting, the present investigation   requires substantial improvement. In my opinion this paper contains nothing new with respect to current literature

General

A revision of the English language by a native speaker must be performed. PLEASE DO CORRECT SPELLING MISTAKES. 

Introduction

·        Some sentences are confused and need to be rephrased

Material and methods

·        An observational study must follow the STROBE guidelines. A STROBE checklist should be attached to the manuscript as supplemental material (www.strobe-statement.org).

·        In the statistical analysis, the authors must indicate how they assessed interobserver variability in measuring the parameters.

·        How the author checked the normality of population?

Discussion

·        Some sentences are confused and need to be rephrased

·        According to the STROBE guidelines the discussion section should give an overview of the literature in the first part and discussing the results that you reached in the second one. 

·        Early loss of correction is a documented complication also after short-segment pedicle screw fixation in treatment of thoracolumbar fracture (PMID: 33679031), the authors should discuss this point.

·        Please specify if the patients wore a T-L brace after the procedure.

Round 2

Reviewer 2 Report

Congratulations to the authors for the review work.